# Xenogenic Implantation of Human Mesenchymal Stromal Cells Using a Novel 3D-Printed Scaffold of PLGA and Graphene Leads to a Significant Increase in Bone Mineralization in a Rat Segmental Femoral Bone Defect

**DOI:** 10.3390/nano13071149

**Published:** 2023-03-23

**Authors:** Steven D. Newby, Chris Forsynth, Austin J. Bow, Shawn E. Bourdo, Man Hung, Joseph Cheever, Ryan Moffat, Andrew J. Gross, Frank W. Licari, Madhu S. Dhar

**Affiliations:** 1Large Animal Regenerative Medicine Program, Department of Large Animal Clinical Sciences, College of Veterinary Medicine, University of Tennessee, Knoxville, TN 37996, USA; 2Department of Biomedical Engineering, University of Tennessee, Knoxville, TN 37996, USA; 3Center for Integrative Nanotechnologies, University of Arkansas at Little Rock, Little Rock, AR 72204, USA; 4College of Dental Medicine, Roseman University of Health Sciences, 10894 S River Front Parkway, South Jordan, UT 84095, USA; 5Department of Orthopedic Surgery Operations, University of Utah, 590 Wakara Way, Salt Lake City, UT 84108, USA; 6Department of Oral and Maxillofacial Surgery, University of Tennessee Graduate School of Medicine, Knoxville, TN 37996, USA

**Keywords:** poly(lactic-co-glycolic acid), additive manufacturing, 3D bioprinting, PLGA blends, rat femur

## Abstract

Tissue-engineering technologies have the potential to provide an effective approach to bone regeneration. Based on the published literature and data from our laboratory, two biomaterial inks containing PLGA and blended with graphene nanoparticles were fabricated. The biomaterial inks consisted of two forms of commercially available PLGA with varying ratios of LA:GA (65:35 and 75:25) and molecular weights of 30,000–107,000. Each of these forms of PLGA was blended with a form containing a 50:50 ratio of LA:GA, resulting in ratios of 50:65 and 50:75, which were subsequently mixed with a 0.05 wt% low-oxygen-functionalized derivative of graphene. Scanning electron microscopy showed interconnected pores in the lattice structures of each scaffold. The cytocompatibility of human ADMSCs transduced with a red fluorescent protein (RFP) was evaluated in vitro. The in vivo biocompatibility and the potential to repair bones were evaluated in a critically sized 5 mm mechanical load-bearing segmental femur defect model in rats. Bone repair was monitored by radiological, histological, and microcomputed tomography methods. The results showed that all of the constructs were biocompatible and did not exhibit any adverse effects. The constructs containing PLGA (50:75)/graphene alone and with hADMSCs demonstrated a significant increase in mineralized tissues within 60 days post-treatment. The percentage of bone volume to total volume from microCT analyses in the rats treated with the PLGA + cells construct showed a 50% new tissue formation, which matched that of a phantom. The microCT results were supported by Von Kossa staining.

## 1. Introduction

Tissue-engineering strategies aim to remodel, replace, or regenerate damaged tissues or organs. Scientists use biodegradable scaffolds in three-dimensional matrices and evaluate the responses of either exogenously implanted MSCs or endogenous/progenitor cells, which are residents in natural tissues. Thus, the key players in these strategies include 3D scaffolds consisting of natural and synthetic polymers and adult MSCs [1,2].

Poly(lactic-co-glycolic acid) (PLGA) is one of the most widely used families of biodegradable and biocompatible polymers in biofabrications. PLGA is a linear copolymer of lactic acid and glycolic acid monomers. It exists in multiple forms with varying ratios of lactic acid (LA) to glycolic acid (GA), resulting in varying molecular weights that ultimately control the rate of degradation. Thus, PLGA can be adapted as a vehicle for the delivery of cells, growth factors, and other molecules in tissue-engineering projects. PLGA is FDA-approved for clinical applications and has suitable mechanical strength to support bone tissue engineering [3,4,5].

Adult, somatic tissue-derived MSCs are extensively used as a characterized source of cells in tissue-engineering projects [6,7,8,9,10,11,12]. Although various sources of stem cells, such as embryonic stem cells (ESCs), MSCs, and induced pluripotent stem cells (iPSCs), have been identified as potential osteoprogenitors for bone tissue engineering, adult MSCs are favored because of their multipotency and immunomodulatory properties. Bone marrow and adipose-derived MSCs have been benchmarked as the most applicable cell sources for bone tissue engineering due to their well-defined in vitro and in vivo osteogenic differentiation patterns [10,11]. Most importantly, the use of MSCs alleviates the need to use ESCs, which is particularly important given the ethical and political concerns associated with the use of ESCs [12].

Reports from several groups, including ours, have shown that nanocomposites containing graphene and its derivatives have varying physical and chemical properties, by virtue of which they can affect cell behavior. We and other groups have shown that graphene-containing nanocomposites can promote adhesion, proliferation, and osteogenic differentiation of MSCs in vitro and in vivo [13,14,15,16,17], making graphene nanoparticles strong candidates for bone tissue engineering.

The combinations of PLGA with commercially available graphene oxide (GO) and reduced graphene oxide (rGO) nanoparticles have been used successfully to demonstrate the enhanced proliferation and osteogenesis of immortalized MC3T3-E1 mouse bone cells and MSCs [18,19]. Many of these studies use nanocomposites containing combinations of PLGA, hydroxyapatite (HA), or bone morphogenetic protein 2 (BMP2) and graphene nanoparticles to induce in vitro osteogenesis, or they fabricate scaffolds using complex material chemistries to promote osteogenesis [20,21]. Similarly, there are in vivo studies using the same components that demonstrate in vivo osteogenesis [20,21]. In these studies, PLGA is combined with another polymer, such as poly(L-lactide) (PLLA), or a bone inductant, such as HA, to induce osteogenesis. Hence, it is likely that the osteogenic effect is primarily due to the osteogenic agents, such as HA or BMP2. Most importantly, we have reported that the topographical features or, in general, the physicochemical properties of the oxidized form of graphene nanoparticles create an osteogenic environment for mesenchymal stem cells in vitro, thus inducing cells to undergo osteogenesis without any chemical induction [15,17]. Hence, the fabrication of an optimal 3D scaffold for in vivo studies is the next step.

The blending of biodegradable polymers is a classical method that has been used for decades to enhance the in vivo performance of implants [22]. Binary blends and composites of biodegradable polymers (PEG, PLLA, PVA, and PLGA) have been suggested to perform better than a single polymer-based scaffold in tissue regeneration [22,23,24]. PLGA has attracted considerable interest as a base material in biomedicine research due to its biocompatibility and tailored biodegradation rate. It has been reported that subtle differences in the molecular weights and LA to GA ratios in a given PLGA sample can affect the release of drugs, small molecules, or even cells, thus presenting an opportunity to control in vivo degradation while serving as a delivery vehicle for in vivo tissue regeneration and repair [3,4,5]. The fabrication of PLGA-based biomimetic scaffolds with specific shapes and properties to modulate cell adhesion, proliferation, and differentiation to enhance bone tissue repair and function poses a significant challenge. Even though PLGA-based blended scaffolds have been proposed for in vitro and in vivo bone regeneration [25,26,27], studies that demonstrate the blending of two PLGA forms of varying molecular weights with carbon nanoparticles are lacking.

In this study, we exploited the properties of PLGA polymers and MSCs by fabricating two novel nanocomposites by blending two forms of PLGA with varying molecular weights and molar ratios of GA and LA. The PLGA blends were mixed with an oxidized form of graphene containing 6–10% oxygen to form a biomaterial ink containing 0.05 wt% graphene. Previously characterized human adipose-derived MSCs (hADMSCs) transduced with a red fluorescent protein (RFP) reporter protein [28] were used to confirm the in vitro cytocompatibility of the scaffolds. Subsequently, one million MSCs were delivered exogenously, and the osteogenic potential was evaluated in vivo using a rat segmental femoral defect model. MicroCT and histological analyses were used to evaluate the responses of the endogenously and exogenously implanted MSCs at 60 days post-treatment.

## 2. Materials and Methods

All biochemicals, cell culture supplements, and disposable tissue culture supplies were purchased from Thermo Fisher Scientific, Waltham, MA, USA unless otherwise stated. In all preparation steps, deionized (DI) water from a Millipore system unit with a resistance of 18 M/cm was used.

### 2.1. Adipose Tissue Collection, Cell Isolation, Characterization, and Transduction

Human adipose tissue was isolated from patients undergoing panniculectomies, in accordance with a protocol approved by the Institutional Review Board at the University of Tennessee Medical Center. All procedures on harvesting the tissues and culturing the MSCs were performed as reported earlier [28,29,30].

Human ADMSCs, previously confirmed to be >80% positive for cluster of differentiation (CD) markers, including CD29, CD44, CD73, and CD90, and negative for CD34 and 45, were expanded in the presence of DMEM/F12, 1% penicillin–streptomycin/amphotericin B, and 10% fetal bovine serum. The cells were grown to 70–80% confluency and then harvested with 0.05% trypsin–EDTA. The cells were cryopreserved in 80% FBS, 10% DMEM/F12, and 10% DMSO, or split and seeded into new flasks for expansion. All experiments were performed using cells from passages 3–4.

### 2.2. Biomaterial Ink and Scaffold Fabrication

The following three forms of PLGA with varying molecular weights and ratios of LA:GA were used in this study:

(1) A ratio of 65:35 with a molecular weight of 40,000–75,000;

(2) A ratio of 75:25 with a molecular weight of 6600–107,000;

(3) A ratio of 50:50 with a molecular weight of 30,000–60,000.

The above three PLGA forms were commercially obtained from Sigma Aldrich, St. Louis, MO, USA.

The functionalized form of the graphene nanoparticles, consisting of 6–10% oxygen, was synthesized. This form of graphene is referred to as low-oxygen graphene (LOG). The synthesis and physicochemical properties of LOG nanoparticles have been described earlier [15,17,31,32]. The physicochemical properties were confirmed to be identical to those reported earlier.

A blend of two molecular weights of PLGA with LOG was used as the biomaterial ink to 3D print the constructs. The two blends of PLGA consisting of equal amounts of 50:50 + 65:35 and 50:50 + 75:25 served as binders for the graphene nanoparticles. These will be referred to as 50:65 and 50:75 scaffolds. The polymeric blends were prepared by mixing 1 g of each form of PLGA with 1 mL of DMSO as the solvent in a 2:1 weight-to-volume ratio. A total of 1 mg of the LOG powder was added to the PLGA blend to give a final concentration of 0.05 wt%. The mixture was continuously rotated in a rotisserie oven for two hours at 65 °C. The mixture was hand-mixed every 15 min to ensure a uniform and complete blending of the PLGA and graphene. Next, the mixture was gas-sterilized using hydrogen peroxide for 28 min, following the protocol described by Sterilis Solutions LLC, MA, USA. The blend was stored in a pneumatic syringe at −20 °C until use.

A commercial 3D bioprinter, Aether 1 (San Francisco, CA, USA), was used to fabricate a consistent and reproducible pattern of the PLGA–graphene blend. A 5 × 5 × 5 mm^3^ scaffold containing a lattice pattern of alternating angles of 0°/45°/90° was fabricated. For the printing, the polymer/graphene blends were removed from the freezer and brought to room temperature before loading the syringe onto the printer. The scaffolds were printed on a sterile tissue culture polystyrene surface with a platform temperature that was maintained at 15 to 30 °C throughout the printing process. The biomaterial ink was extruded at 4–6 bars of pressure with an average rate of 0.5–1.0 mm/s using a 0.2 mm to 0.3 mm inner diameter nozzle, as per a specific G code. It took roughly 2 h to print one scaffold. The printed scaffolds were kept at −20 °C to preserve the designs. Each scaffold was visually evaluated by the investigators while printing and at completion. If visual inspection found it unsatisfactory, the polymer/LOG blend was reheated, and the scaffold was reprinted.

### 2.3. Mechanical and Biological Characterization of Scaffolds

The mechanical analyses consisted of compression of the 5 mm^3^ samples of both iterations using an INSTRON 5965 (Illinois Tool Works Inc., Norwood, MA, USA). The scaffolds were compressed until a 2.5 mm/min deformation was reached [33].

Scanning electron microscopy was used to evaluate the morphology of the biofabricated constructs using standard methods. For the imaging, the samples were fixed in 2% glutaraldehyde at room temperature for 2 h. They were then washed 3 times with 0.1 M phosphate buffer for 10 min each and kept in a 2% osmium solution for 2 h at room temperature in the dark. They were subsequently subjected to an EtOH dehydration sequence. The scaffolds were treated with 30%/50%/70%/80%/90%/96%/100% EtOH each for 15 min, after which they were subsequently placed in a desiccator overnight and imaged.

The human ADMSCs were transduced with RFP for easy visualization on the 3D-printed scaffold using a combination of standard molecular biology techniques [28]. The adhesion, proliferation, and cell-to-cell communication of the transduced cells on the PLGA/LOG scaffolds were monitored for a period of 7–21 days and imaged on a Leica DMi8 microscope. For the in vivo implantation, the scaffolds receiving cells were seeded with one million hADMSCs at least 12–16 h before surgery and maintained in an incubator at 37 °C to ensure that the cells impregnate and attach to the scaffolds.

### 2.4. Animals and Surgery

All animal handling and surgical procedures were conducted under the protocol approved by the University of Tennessee Institutional Animal Care and Use Committee (#2719).

Eight- to twelve-week-old Sprague Dawley rats weighing 22–24 g were used. All rats were housed at a stable temperature of 22 ± 2 °C with a 12 h light/dark cycle and had ad libitum access to drinking water and a standard laboratory rat pellet diet.

To create a 5 mm segmental defect in the femurs, each rat was placed in a lateral position with a randomly selected side facing up. While holding the skin taut and using a sterile blade, a 10–12 mm long incision was made so that the entire femoral shaft was exposed. The fascia was cut, separating the tensor fascia lata and the biceps femoris muscle. The vastus lateralis muscle was elevated from the greater trochanter to the lateral femoral condyle. Two cuts were made in the middle of the diaphysis, and a 5 mm bone segment was cut and removed with a Gigli saw. The presterilized scaffolds (with and without cells) were inserted into the defect and held in place using a 1.1 mm K-wire placed in the intramedullary cavity between the proximal and distal ends of the defect. For 7 days of the post-operative treatment, the rats were given 0.05 mg/kg of buprenorphine SQ every eight to twelve hours and 1µg/mL of enrofloxacin, which was in their drinking water, for pain and infection management, respectively.

The rats were randomly divided into the following 4 treatment groups (*n* = 6 per group): Group 1—PLGA 50:50/65:35 + 0.05 wt% LOG; Group 2—PLGA 50:50/75:25 + 0.05 wt% LOG; Group 3—PLGA 50:50/65:35 + 0.05 wt% LOG + 1 × 10^6^ hADMSCs; Group 4—PLGA 50:50/75:25 + 0.05 wt% LOG + 1 × 10^6^ hADMSCs. The groups represented two acellular and two cellular groups. All of the rats were closely observed for any reactions, general or local, at the site of surgery twice per day for the first week following the operation. At 60 days post-treatment, the animals were sacrificed and their femurs were harvested. The placement of the implant and bone healing were evaluated with conventional radiological analyses, microcomputed tomography (microCT), and histomorphometry.

### 2.5. Radiology and Microcomputed Tomography

A digital radiography system (Philips Easy Diagnost RF System; Cannon DR plates (CXDI-50G); EDR6 Clinical Diagnostic Radiography System) was used. X-rays were taken 12 h post-surgery to confirm the correct K-wire and construct placements and establish a baseline to monitor bone healing. A series of X-rays were conducted at 12 h, 7 days, 30 days, and 60 days post-surgery to assess the formation of newly formed mineralized bone tissues and changes within the segmental bridging.

MicroCT was used to provide insights on the temporal progression of mineralization development during the regenerative process of the femur. For the microCT, all femurs were stored in 10% formalin for 48 h, after which they were placed in 70% ethanol until image scanning. All femurs were scanned using a desktop microCT system (SkyScan 1173; Bruker Kontich, Belgium) at the Roseman University of Health Sciences College of Dental Medicine, UT. They were scanned at 80 kVp with an intensity of 100 μA, resulting in a pixel size of 31.99 μm and 1120 rows × 1120 columns. For the imaging, each femur was placed in a cylinder and attached to a holder with the femur oriented perpendicularly to the image plane.

A 3D image reconstruction of the full femurs was performed using the CT Analyzer software, version 1.20.8.0. (http://www.bruker-microct.com; accessed on 9 April 2021). The region of interest (ROI) was identified visually by drawing polygonal regions for each bone. The ROI included the defect region and a few millimeters of the adjacent, potentially newly formed bone. The image analyses included a bone area/tissue area percentage. Four thresholds ranging between 45–65, 65–85, 85–105, and 105–255 units were selected for the segmentation of the PLGA/graphene nanoparticle construct and the newly formed bone. The 3D models were generated in CTAn and saved in a Standard Triangle Language (STL) file format for image analysis and bone volume measurements.

### 2.6. Histological Staining

The samples (*n* = 6) were embedded in methyl methacrylate, and 5 µm of the undecalcified and calcified sections were obtained (Ratliff Histology Consultants). The undecalcified samples were stained with Von Kossa to visualize the mineralized bones. Von Kossa staining to visualize mineralized bone is a classic and commonly used technique [34,35]. The sections were stained, as reported earlier by our group [15,29,36].

### 2.7. Statistical Analyses

The data were statistically analyzed using a Student’s *t*-test, where *p* < 0.05 was considered as significant.

## 3. Results

### 3.1. Biofabrication of the PLGA–Graphene Nanoparticle Scaffold

The 3D PLGA–graphene nanocomposite scaffolds were fabricated using an Aether 1 3D printer system guided by a custom-generated G code. Two different blends of PLGA + LOG were used as biomaterial inks. The scaffolds were designed using the CAD software Autodesk Fusion 360. The goal was to fabricate a scaffold with a pattern consisting of alternating angles of 45° and 90° with the potential to induce osteogenesis and angiogenesis.

For the biofabrication, a 2D design was laid out to replicate the nozzle path on the stage. A circle with a diameter of 5 mm was formed, and a layer-by-layer build was made to match the rat femur design. Vertical lines for each guideline were laid out in lengths from left to right, 0.30 mm apart, making the first single scaffold layer. Once the single layer was completed, the two constructed prototypes could be made as an assembly of the design. The first layer was secured on the grid and copied; the second layer was then relocated 0.3 mm along the z-axis and rotated 45°. This operation was repeated until the desired geometry was achieved. The final pattern consisted of 15 layers to provide a height of 5 mm. Throughout the printing process, the printer was maintained at a pressure of 5–6 bars, a temperature of 65 °C, an extrusion speed of 0.6 mm/s, and a 0.2 mm nozzle to provide a balance between speed and precision. Scaffolds of 5 mm cubes were generated for in vitro and in vivo applications (Figure 1).

The scanning electron microscope (SEM) provided information associated with the physical characteristics, i.e., topography, spatial distribution of the thread spacing, and porosity of the scaffold. Importantly, the SEM also analyzed the variability in the materials’ composition and structure. The SEM showed complex, yet consistent, convoluted structures, suggesting scaffolds with high surface areas conducive to cell adhesion (Figure 2). The morphological features indicated fusing of the fibers with cracking to increase the peaks and valleys for cell attachment and porosity throughout the scaffolds.

### 3.2. Human Adipose Tissue-Derived MSCs

The human adipose tissue-derived MSCs were isolated and characterized, and primary cultures were generated, as described earlier [28,29]. The cells were characterized with respect to their in vitro trilineage differentiation potential and expression of specific CD markers. As judged by the above results, once the cells were proven to be MSCs [37], they were transduced with RFP to visualize cell adhesion and proliferation.

### 3.3. The PLGA–LOG Scaffold Is Cytocompatible

Attachment of the MSCs to the scaffolds was confirmed by the visualization of red (RFP) fluorescent cells at days 1, 7, 14, and 21 post-seeding (Figure 3). The cells adhered to the scaffolds within 24 h and tended to cluster, with an increase in cellular density between days 1–21. They appeared to increase in density (proliferate) and form clusters, and, with time, the clusters developed communications between each other, confirming that the scaffolds were cytocompatible in vitro.

### 3.4. The PLGA–LOG Scaffold Is Osteoconductive and Osseointegrative

All the rats survived the surgical procedure without any complications. The average weight of each rat increased slightly, indicating good post-operative development and a healthy status throughout the study period. Additionally, no gross or histological evidence of any infection or organ toxicity was observed in any animal. The combination of radiographic imaging and histological analyses demonstrated that the PLGA and PLGA + LOG implants were biocompatible and osteoconductive and did indeed osseointegrate with the host tissues without any adverse effects.

The radiographic images of the femur defects in each rat taken 24 h post-defect ensured that the scaffold and the pin were properly placed into the defect. Subsequently, the images taken throughout the study period confirmed that there were no adverse effects on either the cells or the implants. Most importantly, the scaffolds were able to withstand the weight of the rats, suggesting mechanical integrity. The images taken post-sacrifice showed healing of the defects (Figure 4).

At sacrifice, each implant was harvested and analyzed by microCT to evaluate the percentage bone volume as an endpoint measure of bone healing within the defects. A range of threshold levels was established to evaluate the bone masses at multiple density levels to reduce the difference between cortical bone and trabecular bone and capture the new and mineralized tissues. The analyzed ROI included the defect region and the adjacent, newly formed bone. Each ROI was analyzed at four different thresholds, ranging from 45–255 Hounsfield units. The scans were segmented at each threshold range to capture all new bone and calculate the percentage ratio of bone volume to total volume (%BV/TV) of the implant; this was analyzed with respect to the intact phantom femur. Even though there were no differences in any of the groups when the total range of 45–255 was compared, there were statistically significant differences in specific threshold ranges, indicating differences in the level of mineralized tissues present in each group. For instance, there was a consistently significant increase in response in the animal groups treated with the acellular and cellular 50:75 PLGA + LOG scaffolds in the range of 105–255 units, suggesting a corresponding increase in mineralized tissue (Figure 5). The results showed that the percentage of mineralized tissue in the group treated with 50:75 + hADMSCs was 51.16%, which was the highest and closely matched the value obtained for the phantom (49.97%), providing evidence that this may correspond to the trabecular bone in the ROI. Thus, the data demonstrate the osteoconductive potential of the 0.05 wt% LOG and the stable osseointegration of the 50:75 PLGA + graphene construct with the host tissue. This observation was ultimately confirmed by a positive Von Kossa staining of the mineralized bone tissues in the undecalcified samples. (Figure 5 and Figure 6).

## 4. Discussion

A successful bone tissue-engineering strategy includes the identification and fabrication of an optimal combination of biomaterials/scaffolds and cells [36,37,38,39]. This is achieved by a series of stepwise in vitro and in vivo experiments directed at understanding the physicochemical properties of the biomaterials, the cell response to these features, the fabrication of biomimetic scaffolds by manipulating the materials’ properties and manufacturing processes, and, ultimately, the availability of suitable animal models to test the performance of the biomaterials and cells on the biofabricated scaffolds. In this study, we present data to confirm that a nanocomposite containing a PLGA blend of 50:50 and 75:25 with 0.05 wt% oxidized graphene is a biocompatible, osteoinductive, and osteoconductive platform with safe and efficacious osseointegration with host tissues. To our knowledge, this is the first study to demonstrate the potential of a binary blend of PLGA + LOG constructs for bone tissue engineering.

The in vitro results demonstrated that human ADMSCs adhere, proliferate, and cluster on the scaffolds for up to 21 days (Figure 2). These data support our prior studies and publications from other investigators, demonstrating the cytocompatibility and the osteogenic response of goat and human MSCs on graphene films in vitro [13,14,15,16,17,32]. The cell behavior also confirms that there are no adverse effects when a 99.95% PLGA blend is used as a binder for the LOG nanoparticles to fabricate a 3D scaffold for in vivo applications. Both the 50:65 and 50:75 PLGA blends were cytocompatible. This was not surprising because various forms of PLGA have been used as polymers with graphene nanoparticles in a number of previously reported studies [18,19,20,21]. Our data also confirm the cytocompatibility of the binary PLGA blends with the graphene nanoparticles.

The in vivo methods used in this study and the results obtained are innovative and novel. The novelty begins with the biofabrication method and ends with its application in the rat weight-bearing model. The radiographs, microCT images, histological analyses, and the daily monitoring of the rats confirmed the in vivo cytocompatibility and the osteogenic potential of the PLGA blends and their concentrations and the derivative of the graphene nanoparticles and the xenogenic human MSCs. The cytocompatibility of the LOG nanoparticles and human MSCs supports the data obtained from our two previously reported studies. In 2017, we demonstrated that an LOG with 6–10% oxygen was biocompatible in a rat cortical bone defect model [15]. Similarly, in 2020, we demonstrated that human MSCs expressing CD29, CD44, CD90, and CD105 with the potential to undergo trilineage differentiation are biocompatible and have the potential to heal an alveolar tooth defect in rats [29]. The in vivo biocompatibility and potential osteogenic effects of the PLGA blends in the scaffolds used in this study were novel, yet not unexpected. Due to its biocompatibility and biodegradability, PLGA has been used as an important component for nanocomposite scaffolds in many tissue-engineering studies [3,40,41]. The biological applications of PLGA alone are limited due to its hydrophobic properties, inferior mechanical integrity, and lack of osteogenic activity. As a result, PLGA is often combined with osteogenic inductants, such as HA or BMP2 [20,21]. Hydroxyapatite is an inorganic mineral present in bones and teeth. It plays a role in the structural strength of bones and bone regeneration. Since most of the mineral fractions in bone tissues have an HA structure, synthetic and natural HA and nanoHA are integrated into composite scaffolds for bone repair [36,42,43]. The bone morphogenetic protein 2 is a very extensively characterized growth factor that induces osteogenesis [17,44,45]. Our study is distinct and novel. Based on our previous reports [15,17,32], graphene nanoparticles drive the in vivo osteogenic response in cells, whereas PLGA serves as a binder.

Graphene-based nanomaterials have been recognized as components of bone tissue-engineering scaffolds for many years. Graphene derivatives are preferred over their pristine form and can be produced by the oxidation or functionalization of pristine graphene, with the ultimate goal of reducing pristine graphene’s toxicity and increasing its usability in bone tissue-engineering biomedical applications [46,47,48]. Despite the evident success, there is a lack of uniformity in the manufacturing methods and use, as a result of which the physicochemical properties of graphene nanoparticles might change, resulting in variations in the cell response [49,50,51,52,53,54,55]. The responses can be efficacious or deleterious to cell proliferation and differentiation. These challenges must be overcome prior to their clinical applications. As described in this study, we were able to generate a novel biomaterial ink that consistently produced a reproducible pattern of a 3D-printed scaffold. The scaffold was successfully used in a weight-bearing rat femoral defect model.

Quantitatively, a significant increase in the percentage ratio of bone volume to total volume together with the qualitative analyses of mineralized bone in the femoral defect suggest that both the acellular and cellular nanocomposites of 50:75 PLGA + LOG osseointegrate with the host tissues and illicit an osteogenic response in the cells. The response is higher when the scaffold + cell constructs are used. As suggested in our previous studies, the response is potentially controlled by the physicochemical properties and the topographic features of the graphene nanoparticles and could be a result of scaffold + cell interactions [15,17,32]. In this study, osteogenesis may also be a response due to the specific pattern of the 3D-printed scaffold (Figure 1, Figure 5 and Figure 6). Interestingly, our data support the claim that both the endogenously and exogenously delivered progenitor cells respond to the topographic features, material chemistries, and mechanical properties of the graphene nanocomposite scaffolds [52,53,54,55]. Future experiments to evaluate the signaling mechanisms that trigger osteogenesis can now be initiated. Our data also alleviate the concerns about in vivo nanotoxicity that could be associated with the graphene nanoparticles [56,57].

The major challenges in a bone tissue-engineering project are to identify the optimal biomaterials and then design and fabricate the implant, which can be further complicated by the response or behavior of the progenitor cells towards the implant. This is one of the reasons why an implant may stay on the bench and not get translated into a clinical setting. In this study, we have taken our previous data to the next level by fabricating an optimal bone substitute. We developed a specific G code and fabricated a 3D-printed structure that satisfied all of the criteria for a bone substitute. The combination of a PLGA blend with a biocompatible, osteoinductive, and osteoconductive derivative of graphene along with a 3D biofabrication method provided us with a scaffold with controlled biological and material properties. This provides an osteogenic platform for basic research and clinical applications in bone tissue-engineering projects.

## 5. Conclusions

The advancement of additive manufacturing techniques, such as 3D bioprinting, helps in creating 3D biocompatible implants on which multiple cell types can be seeded. Our study showed that it is possible to successfully fabricate PLGA/carbon nanoparticle nanoscaffolds that exhibit adequate mechanical strength, biodegradation, in vitro and in vivo biocompatibility, and osteogenesis. Within the limits of the study, significant accumulation of mineralized tissue comparable to a phantom, suggesting osteogenesis, was observed in acellular and cellular conditions when a specific PLGA scaffold was implanted into a critically sized rat femur defect. This bone construct offers an equally dynamic combination of osteoconductive three-dimensional structure, osteogenic cells, and osteoinductive growth factors with encouraging mechanical properties. Future studies using the segmental femur defect over longer time points and treated with various concentrations of graphene nanoparticles will hopefully allow for the optimization of a useful PLGA–graphene biofabricated construct as a bone substitute.

## Figures and Tables

**Figure 1 nanomaterials-13-01149-f001:**
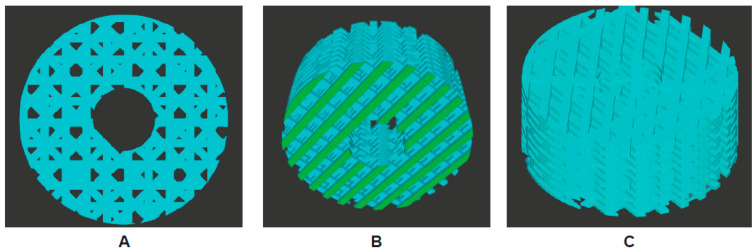
Biofabrication design. Slicer software showing (**A**) front, (**B**) bottom, and (**C**) oblique views of the printed scaffold.

**Figure 2 nanomaterials-13-01149-f002:**
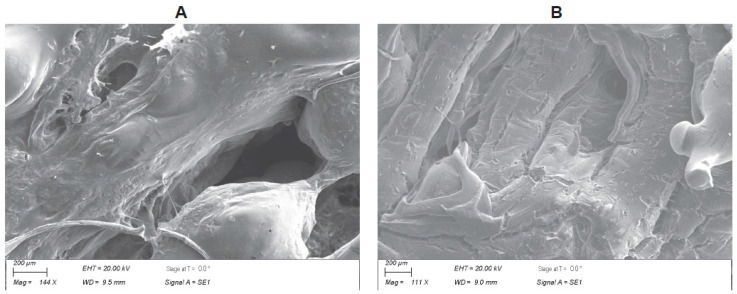
Scanning electron micrograph images of (**A**) a blend of 50:50 and 65:35 poly(lactic-co-glycolic acid) (50:65 PLGA) with 0.05 wt% low-oxygen graphene (LOG) and (**B**) a blend of 50:50 and 75:25 poly(lactic-co-glycolic acid) (50:75 PLGA) with 0.05 wt% low-oxygen graphene (LOG).

**Figure 3 nanomaterials-13-01149-f003:**
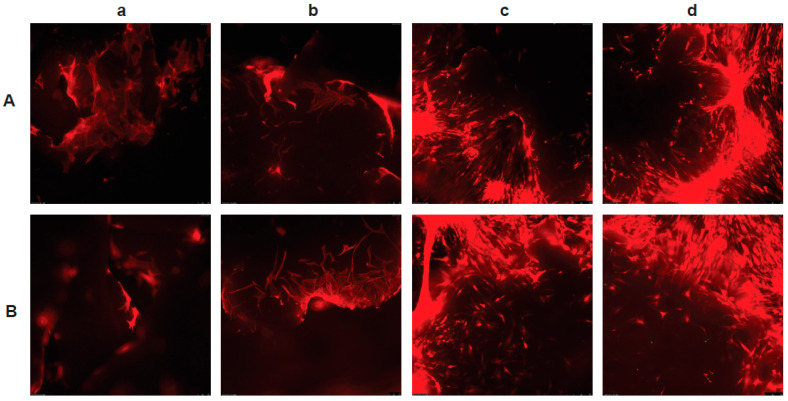
Adhesion and proliferation of hADMSCs. Red fluorescent protein-transduced human ADMSCs adhering, proliferating, and clustering on discrete areas on (**A**) the scaffold of 50:50 and 65:35 poly(lactic-co-glycolic acid) (50:65 PLGA) with 0.05 wt% low-oxygen graphene (LOG) and (**B**) the scaffold of 50:50 and 75:25 poly(lactic-co-glycolic acid) (50:75 PLGA) with 0.05 wt% low-oxygen graphene (LOG) on day 1 (**a**), day 7 (**b**), day 14 (**c**), and day 21 (**d**) post-seeding. On day 1, adherence was noted, followed by proliferation (indicated by an increase in cell density) and cell-to-cell communication (indicated by cell clustering) on days 7, 14, and 21.

**Figure 4 nanomaterials-13-01149-f004:**
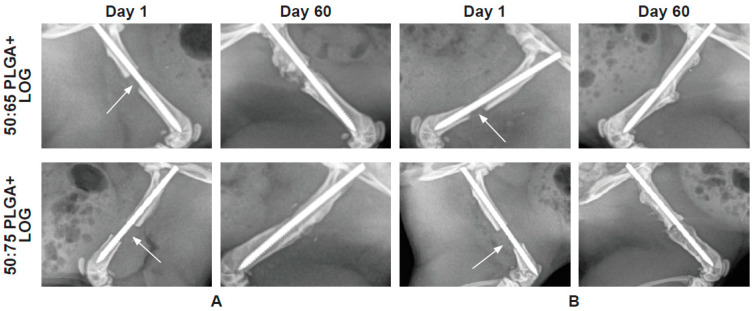
Radiography. Representative images of the rat femoral defects treated with the two scaffolds of poly(lactic-co-glycolic acid and low oxygen graphene (PLGA + LOG) alone ((**A**)—acellular) and with 1 × 106 hAD-MSCs ((**B**)—cellular) at days 1 and 60.The arrows show the defect at day 1, which is filled by day 60. The scaffolds are radiopaque and hence are not visible in the radiographs.

**Figure 5 nanomaterials-13-01149-f005:**
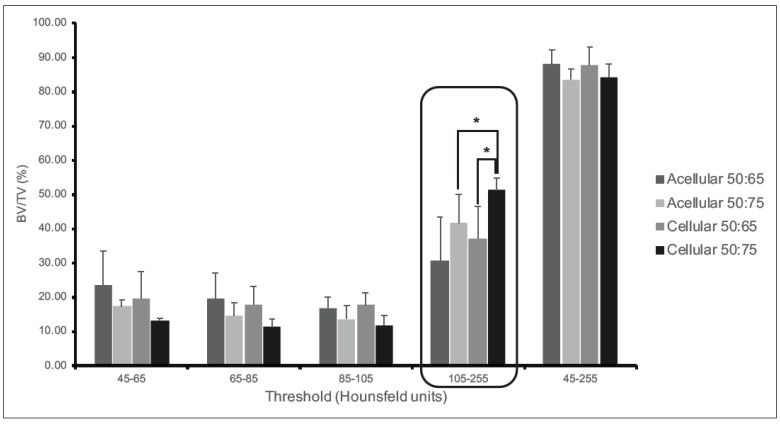
MicroCT data analysis. Graphical representation of percent bone volume/total volume (%BV/TV) from the microCT scans of all the rat femurs. Data is reported as a function of individua threshold Hounsfeld units. As described in the Section 3 these data are significant in the 105–255 unit range, which denotes 50% new bone and, which is closely matched to that of the phantom. Asterisks denote significant increase in the %BV/TV in rats treated with 50:75 + MSCs compared to those treated with 50:75 alone. Similarly, there is a significant increase in %BV/TV in rats treated with 50:75 + MSCs compared to those treated with 50:65 + MSCS.

**Figure 6 nanomaterials-13-01149-f006:**
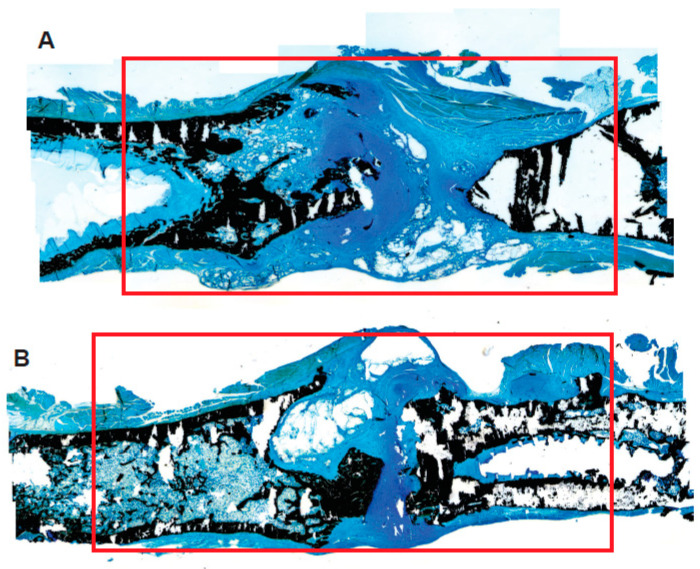
Histological analyses. Representative images showing the Von Kossa staining of the region of interest (ROl) for a rat treated with (**A**) acellular i.e., PLGA 50:75 scaffold alone, and (**B**) PLGA 50:75 + 1 million hAD-Mscs. The RO is shown as a red box. Note the black staining indicating new mineralized tissue in the defect. Note that the medullary cavity as visualized with the screw marks. is filled with new bone in (**B**).

## Data Availability

All data that were generated or analyzed during this study are included in the article.

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
