# Peer review of "Xenogenic Implantation of Human Mesenchymal Stromal Cells Using a Novel 3D-Printed Scaffold of PLGA and Graphene Leads to a Significant Increase in Bone Mineralization in a Rat Segmental Femoral Bone Defect"

_nanomaterials, 2023, doi:10.3390/nano13071149_

Round 1

Reviewer 1 Report

 Xenogenic implantation of human mesenchymal stromal cells 2 using a novel 3D printed scaffold of PLGA and graphene leads 3 to a significant increase in bone mineralization in a rat seg-4 mental femoral bone defect

Dear Editor,

This paper presents interesting results which can be useful for tissue-engineering but the way of presentation should be better. First basic remark: I couldn't properly review the paper because there were no drawings in it, and I couldn't find an additional file with drawings, so I reviewed only the text of the manuscript.

This paper can be recommended to publish after revision and in my opinion the following changes should be done before the paper acceptance (yellow marked in the file).

Abstract

Too long sentences

Keywords

should be more precise

Introduction

The introduction part should be extended by other references describing e.g. hyaluronic acid or other polysaccharide. For example:   

Characteristics of hybrid chitosan/phospholipid-sterol, peptide coatings on plasma activated PEEK polymer, Materials Science and Engineering: C, 120 (2021) 111658

The effect of chitosan/TiO2/hyaluronic acid subphase on the behaviour of 1,2-dioleoyl-sn-glycero-3-phosphocholine membrane, Biomaterials Advances, 2022, 138, 212934 

Line 234

a.      stl please explain shortcut

Undecalcified samples were stained with Von Kossa to visualize mineralized bone, give citation or explain???

Line 268-271 Line 330 and others , Line 361

Line 278-280;  Line 309-321 many repetitions???

Line 288

End of sentence, should be dot

Line 425

In situ should be italics

Correct English style

Please specify the exact purity of all ingredients.

Figure 5

In the figure, some fragments are written in large, others in small type, you can also see the difference in font size.

References

First of all, the literature list should be standardize, sometimes should be added full data.  Please add new positions about other polymers ideal for tissue engineering. For example:

Wetting Properties of Polyetheretherketone Plasma Activated and Biocoated Surfaces, Colloids and Interfaces  3(1), 40 (2019) 1-14

Some old proposition should be removed from list of references

I can recommend this article, but after revision.

Reviewer 2 Report

This paper reports on the development of a new bone substitute based on the 3D printing of biomaterial inks made of poly (lactic-co-glycolic acid) loaded with 0.05 wt% low oxygen graphene nanoparticles. The methodology is well described and the results are promising. Nevertheless, the impact of this work could be amplified by the following improvements:   

1. The Discussion section needs more thorough referencing. A vast bibliography on top-down tissue engineering proves the biocompatibility and biodegradability of PLGA (see, e.g. [Lanza et al. 2007. Principles of Tissue Engineering, 3rd Edition, Burlington: Elsevier Academic Press]). Therefore, the sentence "The cell behavior suggests the lack of adverse effect from PLGA." sounds like reinventing the wheel. Also, on line 376, the authors mention a "growing number of studies" without citing any of them.  In an effective discussion, the reported results are contrasted with previous findings. Collective citations such as (14, 16, 39) - line 348 - do not help the reader in this respect. Please revise Section 4 by citing the proper literature and highlighting the originality of the present work. 

2. Part of the terminology of this manuscript needs to be updated as follows: 

a. In recent literature, bioink is defined as a 3D printable, biocompatible, cell-laden material [Groll et al. 2018. A definition of bioinks and their distinction from biomaterial inks. Biofabrication 11:013001; DOI: 10.1088/1758-5090/aaec52]. Hence, on line 247, instead of "the Bioinks" a more appropriate expression would be "biomaterial inks". Please replace "bioink" with "biomaterial ink" throughout the entire text (lines 24, 25, 103, 128, 138, 155, 247).   

b. The term "biofabrication" is wider than "bioprinting"  [Groll et al. 2016. Biofabrication: reappraising the definition of an evolving field. Biofabrication 8:013001; DOI: 10.1088/1758-5090/8/1/013001]. Therefore, I would not include it in the family additive manufacturing techniques (only bioprinting is part of it). Thus, on line 385, instead of "3D Biofabrication/Bioprinting" I would write "3D bioprinting".

3. Minor revisions: 

Line 349: A reference is needed in the sentence "Additionally, prior 349 studies suggest ...". 

Line 359: Instead of "reasons, why" I would write "reasons why". 

Line 365: I would remove the new paragraph indentation from lines 366 and 370. 

Line 374: I would delete "The use of ... enabled us to complete this project successfully." It is a vague statement with little information content.  

Figures need to be formatted according to the Instructions for Authors ( https://www.mdpi.com/journal/nanomaterials/instructions): label of multi-panel figures as (a), (b), (c) etc., and use the  "MDPI_5.1_figure_caption" style for the figure title and caption. 

Reviewer 3 Report

1. In line 240, it should be 'Student's t-test.' There are many such mistakes. The authors should check the manuscript thoroughly.  

2. In Fig. 2, provide the elemental mapping. Also, expand the abbreviations in the figure caption.

3. In Fig. 3, also provide the quantitative analysis.

4. In Fig. 4, label the figure where are defects, scaffolds, etc. 

5. In Fig. 5, Expand the abbreviations in the figure caption.

6.  In Fig. 7, label the figures where are screws mark, medullary cavity, etc.

7. The quality of all figures is very low; authors should put good-resolution figures. Also, the caption of the figures should be put in text format, not image format. 

Round 2

Reviewer 1 Report

My comments were not taken into account.

Reviewer 2 Report

The revised manuscript addresses most of my concerns related to the original text. The terminology has been improved, and formatting issues have been clarified. The results are promising and well-documented, so I see no technical reasons to reject this work.

Nevertheless, the authors might wish to improve Section 4 (Discussion). Certain aspects have been clarified, but literature citations are still scarce, mainly collective (i.e. many papers cited in a single sentence), and loosely connected to the original findings reported here. 

Reviewer 3 Report

Authors have addressed most of my comments. However, still, the response to the second comment that is regarding elemental mapping is not satisfactory. Also, the cited references don't contain elemental mapping, though in ref. 15, it's elemental analysis, not mapping.

Authors should do it before publishing. 

Round 3

Reviewer 1 Report

I've been reviewing work for about 20 years
and this is the first time this has happened to me.

Reviewer 3 Report

I am wondering how the spatial distribution of graphene will not affect the property of the PLGA scaffold. There are high chances of aggregation of graphene. Thus, authors are suggested to provide the elemental mapping.